# A Proinflammatory Diet May Increase Mortality Risk in Patients with Diabetes Mellitus

**DOI:** 10.3390/nu14102011

**Published:** 2022-05-11

**Authors:** Jiaxing Tan, Nuozhou Liu, Peiyan Sun, Yi Tang, Wei Qin

**Affiliations:** 1Division of Nephrology, Department of Medicine, West China Hospital, Sichuan University, Guoxuexiang Street, Chengdu 610041, China; xingest@foxmail.com (J.T.); tmka1986@163.com (Y.T.); 2West China School of Medicine, Sichuan University, Chengdu 610041, China; liunuozhou@stu.scu.edu.cn (N.L.); spy200212@163.com (P.S.)

**Keywords:** dietary inflammatory index, diabetes mellitus, all-cause mortality, National Death Index, NHANES

## Abstract

This was an observational study based on the National Health and Nutrition Examination Survey (NHANES) and National Death Index (NDI) 2009–2014 which aimed to validate whether a proinflammatory diet may increase mortality risk in patients with diabetes mellitus. Dietary inflammatory potential was assessed by dietary inflammatory index (DII) based on 24 h dietary recall. Mortality follow-up information was accessed from NDI, which was then merged with NHANES data following the National Center for Health Statistics (NCHS) protocols. For 15,291 participants from the general population, the average DII was 0.37 ± 1.76 and the prevalence rate of diabetes was 13.26%. DII was positively associated with fasting glucose (β = 0.83, 95% CI: 0.30, 1.36, *p* = 0.0022), glycohemoglobin (β = 0.02, 95% CI: 0.01, 0.03, *p* = 0.0009), and the risk of diabetes (OR = 1.05, 95% CI: 1.01, 1.09, *p* = 0.0139). For 1904 participants with diabetes and a median follow-up of 45 person-months, a total of 178 participants with diabetes died from all causes (mortality rate = 9.34%). People with diabetes who adhered to a proinflammatory diet showed a higher risk of all-cause mortality (HR = 1.71, 95%CI: 1.13, 2.58, *p* = 0.0108). In summary, DII was positively associated with diabetes prevalence and a proinflammatory diet may increase mortality risk in patients with diabetes mellitus.

## 1. Introduction

Diabetes mellitus (DM) is a metabolic condition characterized by chronic hyperglycemia which may lead to tissue or organ inflammation and damage [1]. According to the International Diabetes Federation (IDF), more than 536 million individuals aged between 20 and 70 suffer from diabetes all over the world, causing an estimated annual global health expenditure of USD 673 billion. DM has become a serious public health concern [2].

Emerging studies have reported that inflammation plays a key role in the pathogenesis of DM [3]. It has been demonstrated that patients with diabetes tend to have a higher level of inflammatory cytokines, such as tumor necrosis factor α (TNF-α) and interleukin (IL-6). TNF-α can inhabit insulin signaling by stimulating serine/threonine phosphorylation of Irs-1/Irs-2, thus inhibiting insulin/IGF-stimulated tyrosine phosphorylation and corresponding signal transduction [4]. Similarly, IL-6 also exerts a strong negative influence on insulin signal transduction, which may aggravate DM [5].

Solid evidence indicates that diet may effectively regulate the body’s inflammatory state [6]. Paralleling these studies has been an attempt to quantitatively measure the inflammatory potential of diet. Thus, the dietary inflammatory index (DII) is widely used to give a comprehensive score for whether a diet is anti- or proinflammatory [7]. It has been confirmed that DII is closely related to some key biomarkers in inflammation and has something to do with a variety of noncommunicable diseases [8]. However, few studies have investigated the relationship between DII and DM directly. In this study, we used data from the National Health and Nutrition Examination Survey (NHANES), a database on health and nutritional status among the US population [9]. The primary aim of this study was to assess the association between DII and mortality among people with diabetes, which could shed new light on a variety of fields, including dietary management, nutritional epidemiology, and diabetes care.

## 2. Materials and Methods

### 2.1. Participant Recruitment

NHANES is an ongoing and nationwide health survey managed by the National Center for Health Statistics (NCHS) at the U.S. Centers for Disease Control and Prevention (CDC). A complex and multistage probability design with representative sample weights was applied in the NHANES database to accurately estimate the nutritional and health status of noninstitutionalized U.S. civilians. The NHANES database contained five different types of data, including Demographics, Dietary, Examination, Laboratory, and Questionnaire information. All data in our study are publicly available at https://www.cdc.gov/nchs/nhanes/ (accessed on 1 January 2022). The survival condition of participants in NHANES was collected in the National Death Index (NDI) and was not available for public release in the cases of participants below 18 years old [10].

We combined NHANES and NDI 2009–2014 and selected 17,862 participants eligible for both NHANES and NDI data. Then, a total of 1387 participants were excluded for incomplete DII calculation data. We also excluded participants whose data were incapable of defining diabetes (see Exposure and Outcome Definitions) (*n* = 1184). Finally, 15,291 participants with full data for DII and diabetes were utilized to assess the relationship between DII and diabetes prevalence. In addition, 1904 participants with diabetes were used to assess the association between DII and all-cause mortality among people with diabetes. The process of sample collection is summarized in Figure 1.

### 2.2. Exposure and Outcome Definitions

We calculated DII from 24 h dietary recall data in the NHANES “Questionnaire” section which was validated by the Nutrition Methodology Working Group and strictly followed the calculation protocols presented by Nitin Shivappa et al. [7,11]. A total of 28 food parameters were used in the DII calculation, including alcohol, caffeine, protein, fiber, β-carotene, cholesterol, carbohydrates, energy, fats, n-3 fatty acids, n-6 fatty acids, poly-unsaturated fatty acids, mono-unsaturated fatty acids, saturated fat, thiamin, magnesium, zinc, selenium, iron, riboflavin, folic acid, vitamin A, vitamin B-6, vitamin B-12, vitamin C, vitamin D, vitamin E, and niacin. Previous studies have illustrated that DII calculations with fewer than 30 food parameters do not affect predictability [12]. Diabetes was defined as a self-reported diabetes diagnosis, the use of oral hypoglycemic agents or insulin, HbAlc ≥ 6.5%, a plasma glucose level ≥ 200 mg/dL at 2 h after the oral glucose tolerance test (OGTT), or a fasting glucose level ≥ 126 mg/dL [13]. Since NHANES did not directly document mortality data, mortality information in this study was based on a probabilistic match between NHANES and NDI following procedures validated by the NCHS [14]. Mortality information in the NDI was recorded through 31 December 2015.

### 2.3. Covariates

Continuous variables included age, energy and protein intake, systolic and diastolic pressure, estimated glomerular filtration rate (eGFR), and urinary albumin-to-creatinine ratio (UACR). Categorial variables included sex, race, hypertension, chronic kidney disease (CKD), physical activity, smoking exposure, alcohol intake, and body mass index (BMI). The eGFR was calculated from the serum creatinine level through the CKD-EPI formula [15]. The measurement of UACR from spot urine samples was suggested by “Kidney Disease: Improving Global Outcomes” (KDIGO), where UACR > 30 mg/g was defined as “albuminuria” [16]. KDIGO defined CKD as the existence of either eGFR < 60 mL/min/1.73 m^2^, markers of kidney damage (e.g., albuminuria), or both, of at least 3 months duration, regardless of the underlying cause [17]. CKD was suggested to be classified into 5 stages by KDIGO: CKD stage 1: eGFR ≥ 90 mL/min/1.73 m^2^; stage 2: 60 mL/min/1.73 m^2^ ≤ eGFR < 90 mL/min/1.73 m^2^; stage 3: 30 mL/min/1.73 m^2^ ≤ eGFR < 60 mL/min/1.73 m^2^; stage 4: 15 mL/min/1.73 m^2^ ≤ eGFR < 30 mL/min/1.73 m^2^; and stage 5: eGFR < 15 mL/min/1.73 m^2^. Hypertension was defined based on a systolic blood pressure ≥ 140 mmHg, a diastolic blood pressure ≥ 90 mmHg, or a self-reported hypertension diagnosis [18]. Physical activity was divided into three groups from the NHANES questionnaire data according to a previous study, including vigorous, moderate, and less than moderate activity [19]. We also divided participants into three groups according to their BMI, including normal (BMI < 25 kg/m^2^), overweight (25 ≤ BMI ≤ 30 kg/m^2^) and obese (BMI > 30 kg/m^2^) [20]. Alcohol intake was also defined from the NHANES questionnaire according to a previous study [21]. Participants who consumed <12 drinks in their entire life were grouped as non-drinkers. Those who reported ≥12 drinks consumed in any year or their entire life but not in the past year were categorized as former drinkers. Those who reported ≥12 drinks consumed in any year or their entire life and one or more drinks in the past year were categorized as current drinkers. To reduce recall bias, we defined smoking exposure from the serum cotinine level instead of the “cigarette-use questionnaire”, where serum cotinine > 10 ng/mg for current smokers, 0.011 ≤ serum cotinine ≤ 10 ng/mg for former smokers, and serum cotinine < 0.011 ng/mg for non-smokers [22,23].

### 2.4. Statistical Analysis

We followed CDC analytical guidelines when performing all statistical analyses, where an appropriate sample weight was also applied to each participant due to the NHANES complex multistage cluster survey design [22]. Categorial variables are presented as proportions, while continuous variables are presented as the mean ± standard deviation (SD). A weighted chi-square test for categorical variables or a *t*-test for continuous variables was employed to analyze differences between people with DII > 0 and DII < 0. The relationships between DII and fasting glucose and glycohemoglobin were assessed by multivariable linear regressions, while those between DII and diabetes prevalence were evaluated by multivariable logistic regressions. To represent hierarchal adjustment for regression models, three different models were used (Model 1, unadjusted; Model 2, adjusted for age, gender, and race; Model 3, adjusted for age, gender, race, energy intake, protein intake, systolic pressure, diastolic pressure, smoking exposure, alcohol intake, and BMI). When assessing the association between DII and mortality among people with diabetes, we performed univariate and multivariate Cox regression independently and presented the results as hazard ratios (HRs), 95% confidence intervals (95%CIs), and *p*-values. DII, age, sex, race, energy and protein intake, eGFR, UACR, systolic and diastolic pressure, physical activity, smoking exposure, alcohol intake, and BMI were included in multivariate Cox regression. All analyses were based on R version 4.0.5 (http://www.R-project.org (accessed on 1 January 2022), The R Foundation).

## 3. Results

### 3.1. The Association between DII and Diabetes Prevalence

The baseline characteristics of the participants are summarized in Table 1. A total of 15,291 participants were enrolled in this study, with an average age of 47 ± 18 years, of whom 49% were male and 51% were female. The mean DII was 0.2 ± 1.8, whereas 7914 participants were grouped in a proinflammatory diet (DII > 0). Participants adhering to a proinflammatory diet tended to have lower energy (1719 ± 674 vs. 2610 ± 1044 kcal, *p* < 0.0001) and protein intake (64 ± 29 vs. 102 ± 45 g, *p* < 0.0001) relative to those with DII < 0 (anti-inflammatory diet). We also identified higher glycohemoglobin, eGFR, and UACR in people with a proinflammatory diet, while there was no material difference among systolic pressure and fasting glucose. The observed differences in plasma concentrations of fasting glucose and glycohemoglobin were too small to be considered clinically relevant. The overall prevalence rates of hypertension, diabetes, and CKD were 17.2%, 13.3%, and 11.5%, respectively, and people with DII > 0 seemed to have a higher risk of diabetes (11.4 vs. 8.9%, *p* < 0.0001) and CKD (10.5 vs. 7.7%, *p* < 0.0001) compared with those with DII < 0. There was no difference in prevalence of hypertension (*p* = 0.1191). People with a proinflammatory diet were more likely to have a lower level of physical activity and to be current smokers, current drinkers, or obese.

Table 2 presents the results of regression analyses evaluating the association between DII and diabetes. In the unadjusted model (Model 1), we identified positive associations between DII and fasting glucose (β = 0.45, 95% CI: 0.06, 0.84, *p* = 0.0236), glycohemoglobin (β = 0.02, 95% CI: 0.02, 0.03, *p* < 0.0001), and risk of diabetes (OR = 1.07, 95% CI: 1.05, 1.10, *p* < 0.0001). These associations became more evident after we divided people into two continuous groups (DII < 0 and DII > 0). These results remained stable when we adjusted for age, sex, and race. After we adjusted for all potential covariates (Model 3), we still found that people with higher DII showed higher fasting glucose and glycohemoglobin levels and an increased likelihood of diabetes. However, the association between DII and risk of CKD did not remain significant after we adjusted for all covariates. For people with DII > 0 (proinflammatory diet), a higher fasting glucose level (β = 1.91, 95% CI: 0.16, 3.66, *p* = 0.0323) and an increased risk of diabetes (OR = 1.18, 95% CI: 1.03, 1.34, *p* = 0.0141) and all-cause mortality (OR = 1.26, 95% CI: 1.02, 1.57, *p* = 0.0357) were observed relative to those with DII < 0 (anti-inflammatory diet). However, the numerical differences in both fasting glucose and glycohemoglobin were too small to matter to a clinician or a patient.

### 3.2. The Association between DII and All-Cause Mortality among People with Diabetes

The sociodemographic and dietary characteristics of participants with diabetes were stratified by DII > 0 and DII < 0 and are presented in Table 3. A total of 1904 participants with a mean person-month follow-up of 47.77 ± 21.38 were evaluated. The average age and DII were 60 ± 14 years old and 0.4 ± 1.8, respectively. Among them, 54% were male and 46% were female. We did not find a difference in eGFR, UACR, glycohemoglobin, fasting glucose, or systolic pressure between people with DII > 0 and those with DII < 0. The prevalence of CKD was serious (prevalence rate = 29.5%), and people with a proinflammatory diet (DII > 0) showed a higher risk of stage 1 or 2 CKD (9.6 vs. 7.5%; 18.4 vs. 14.6%, *p* < 0.0001) relative to those with DII < 0. About 30% people had diabetes combined with hypertension; however, the prevalence of hypertension was not different between DII < 0 and DII > 0 (*p* = 0.8896). Less than 20% of people with diabetes reported vigorous physical activity, and more than half of those people were second-hand smokers, current drinkers, or obese. The all-cause mortality rate was 9.35%, and people with DII > 0 showed a higher mortality rate than those with DII < 0 (9.47 vs. 5.84%, *p* < 0.0001).

During a median follow-up of 45 person-months, a total of 178 participants with diabetes died from all-causes. The Kaplan–Meier survival curve for all-cause mortality stratified by DII > 0 and DII < 0 is shown in Figure 2. It was found that mortality was higher in individuals with a proinflammatory diet than in those with an anti-inflammatory diet (log-rank test; *p* = 0.014).

The results of Cox regression analysis are summarized in Table 4. In univariate analysis, we found that DII > 0 (proinflammatory diet), non-Hispanic white ethnicity, older age, and higher systolic pressure might be positively associated with the all-cause mortality rate. By comparison of those with less than moderate physical activity and those with vigorous (HR = 0.40, 95% CI: 0.24, 0.66, *p* = 0.0003) or moderate physical activity (HR = 0.57, 95% CI: 0.41, 0.80, *p* = 0.0011), the latter seemed to be a protective factor. Similar results were also identified among people who were non-smokers or second-hand smokers relative to current smokers. In multivariate analysis, people adhering to a proinflammatory diet (DII > 0) had 71% increased all-cause mortality (HR = 1.71, 95% CI: 1.13, 2.58, *p* = 0.0108). Age was still positively associated with all-cause mortality (HR = 1.07, 95% CI: 1.04, 1.09, *p* < 0.0001), while non-smokers (HR = 0.06, 95% CI: 0.04, 0.11, *p* < 0.0001) and second-hand smokers (HR = 0.39, 95% CI: 0.24, 0.66, *p* = 0.0003) tended to have lower all-cause mortality rates than current smokers.

## 4. Discussion

The DII is a literature-derived and population-based index that was developed to robustly compare the likelihood of inflammation in the diets of different populations. A DII greater than 0 indicates that the diet of individuals tends to be proinflammatory and vice versa [7]. An increasing number of studies have reported that diet and the body’s inflammatory state are closely linked, and higher DII scores are associated with the prevalence and severity of major noncommunicable diseases [8]. However, few studies have investigated the relationship between DII and diabetes. Hence, this observational study enrolled 15,291 participants in total to assess the correlation between DII and all-cause mortality among people with diabetes. Our results revealed that DII was positively associated with diabetes prevalence in the general U.S. population and that people with diabetes and adhering to a proinflammatory diet (DII > 0) had a higher all-cause mortality rate and lower survival probability relative to those with an anti-inflammatory diet (DII < 0).

Specifically, DII was found to be positively related to the serum levels of fasting glucose and glycohemoglobin that could reflect the severity of DM and whether DM was well controlled. It was indicated that participants with a proinflammatory diet were more likely to suffer from DM, implying that an inflammatory diet might be involved in the development of diabetes [24]. Recent epidemiological studies have proposed that inflammation is closely associated with DM and may participate in its pathophysiological derangements [3,25]. It has been found that adipose tissue appears to be blamed since it may be the main site of production of inflammatory cytokines, which explains why obese people tend to suffer from insulin resistance and DM [3,25,26]. This view was also confirmed in our study. Our study revealed that patients with a proinflammatory diet had higher BMIs and obesity rates, indicating that proinflammatory diets were more likely to lead to obesity, which might induce an immunological–metabolic crosstalk disorder and cause the development of diabetes.

In addition to being strongly associated with the development of diabetes, a proinflammatory diet was also associated with poor prognosis. Both the Kaplan–Meier survival curve and Cox regression analysis in our study showed that mortality was significantly higher in diabetic patients who had a proinflammatory diet. A total of 28 food parameters are used in the DII calculation, including alcohol, caffeine, protein, fiber, β-carotene, cholesterol, carbohydrates, energy, fats, n-3 fatty acids, n-6 fatty acids, poly-unsaturated fatty acids, mono-unsaturated fatty acids, saturated fat, thiamin, magnesium, zinc, selenium, iron, riboflavin, folic acid, vitamin A, vitamin B-6, vitamin B-12, vitamin C, vitamin D, vitamin E, and niacin, which are the nutrients that are commonly found in our daily life [7]. It was suggested that DII should be used clinically to evaluate the diet of diabetic patients, and physicians should advise patients to choose an anti-inflammatory diet. To the best of our knowledge, this was the first clinical trial to investigate the correlation between mortality and DII in diabetic patients. Notably, it has been proven to be beneficial for DM patients to eat a Mediterranean diet, a vegetarian diet, a traditional Korean diet, a Japanese diet, and antihypertensive and low-glycemic index diets [27,28]. Esposito et al., in 25 patients with metabolic syndrome, found that high-fat meals (pro-inflammatory diet) could increase TNF-α levels, which was further associated with endothelial dysfunction [29]. Although our study cannot answer which type of diet is most beneficial for people with diabetes, the above-mentioned diets have been reported to have some anti-inflammatory effects, providing further evidence that a low-DII diet may be recommended for people with diabetes.

It is worth mentioning that the molecular mechanism by which DII is related to the prognosis of diabetes is not fully understood at present. The DII was scored based on whether each dietary parameter had effects on six common cytokines, including high-sensitivity C-reactive protein (hs-CRP), IL-1β, IL-4, IL-6, IL-10, and TNF-α [7]. Among these, IL-1β, IL-6, TNF-α, and hs-CRP have been proven to be proinflammatory, while IL-4 and IL-10 showed significant anti-inflammatory effects [30]. Several further biochemical studies have demonstrated the networks in which different molecules, including insulin, cytokines, and their corresponding receptors, cooperate with each other to regulate the metabolic process. For example, TNF-α can activate c-Jun N-terminal kinase (JNK) and inhibit the phosphorylation of Irs-1/Irs-2 [31]. As a result, the insulin signaling pathway is disrupted, which inhibits the regulatory effect of insulin. Similar to this, after combining with corresponding cytokines, such as IL-2, receptors can stimulate cytokine signaling-1 (SOCS1) and cytokine signaling-3 (SOCS3) through a certain pathway. SOCS1/3 can inhibit the phosphorylation of Irs-1/Irs-2, similar to JNK [31], inhibiting the regulatory effect of insulin. In contrast, anti-inflammatory cytokines show protective effects in DM patients. IL-4 can induce regulatory T cells and then reduce the severity of inflammation through complex cell interactions [32]. This evidence indirectly proves that DII might be related to the severity and prognosis of DM.

Our study had several strengths. To the best of our knowledge, this was the first large-scale observational study to assess the relationship between DII and diabetes prevalence and all-cause mortality among representative samples with DII from the U.S. population. This research was expected to shed new light on several important public health fields, such as dietary management, nutritional epidemiology, and diabetes care. Furthermore, the potential covariates in this study were generally wide and meaningful. Not only were laboratory covariates such as systolic pressure, diastolic pressure, and UACR included, essential covariates in health-related life habits, such as alcohol intake, smoking exposure, and physical activity, were also incorporated. In terms of the research methods, we used serum cotinine to define smoking exposure instead of the “cigarette-use questionnaire”, which could reduce the recall bias to a great extent. However, some limitations still existed in our study inevitably. First, due to the limitation of the cross-sectional design, no causal relationship could be established between DII and prevalence of diabetes mellitus. Association does not imply a cause-and-effect relationship. Prospective longitudinal investigation will be required to explore that matter. Second, recall bias still inevitably exists despite the efforts we made (e.g., we utilized serum cotinine levels to define smoking exposure instead of the smoking questionnaire). The calculation of DII totally based on 24 h dietary recall and the definition of alcohol intake as well as physical activity were only collected from self-reported data, whose accuracy were, to some extent, uncertain. Moreover, the duration of the self-reported diet for each individual was not clear in the NHANES database. As for the results of the Cox analysis, we could not obtain the internal relationship between comorbid factors (covariates) on mortality. Lastly, there might be other potential variables that could have had unknown impacts on our results. 

## 5. Conclusions

The DII was positively associated with diabetes prevalence and a proinflammatory diet may increase mortality risk in patients with diabetes mellitus. Therefore, the utilization of the DII in assessing dietary patterns might be very useful in future public health management. However, more clinical trials are needed to help clarify the exact effect of a proinflammatory diet.

## Figures and Tables

**Figure 1 nutrients-14-02011-f001:**
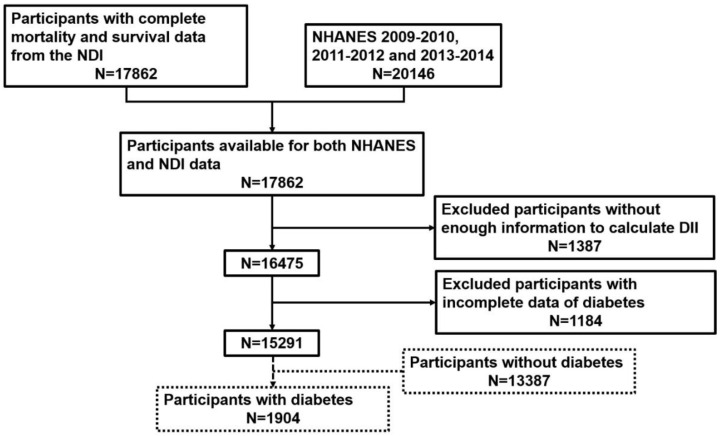
Flow chart of individual inclusion and exclusion.

**Figure 2 nutrients-14-02011-f002:**
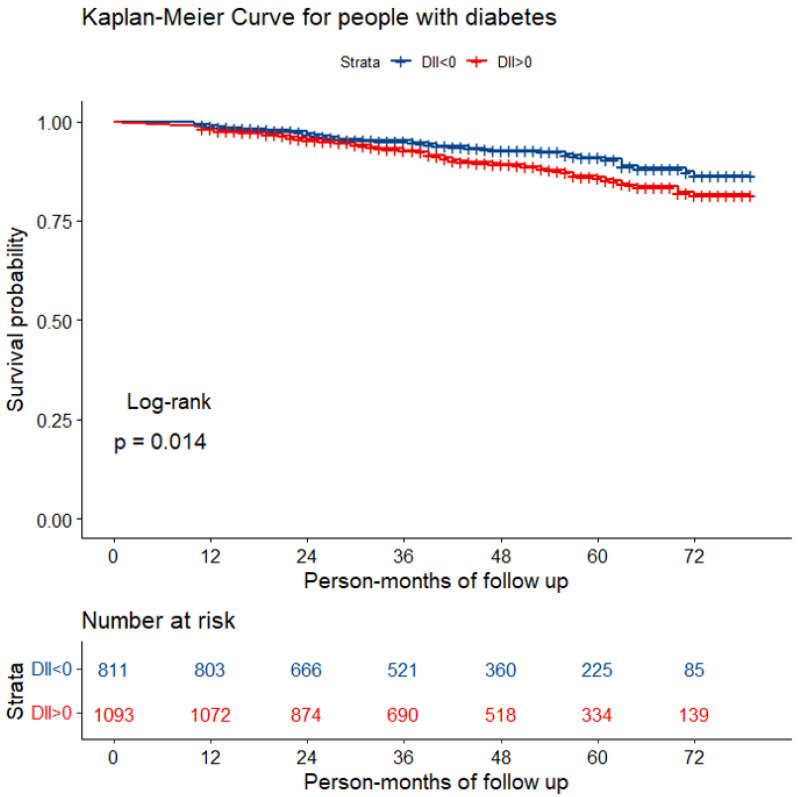
The Kaplan–Meier curve for diabetic patients with different diets.

**Table 1 nutrients-14-02011-t001:** Baseline characteristics of participants from NHANES 2009–2010, 2011–2012, and 2013–2014.

Dietary Inflammatory Index	Overall	Anti-Inflammatory Diet (DII < 0)	Proinflammatory Diet (DII > 0)	*p*-Value
Participant number	15,291	7377	7914	-
Mean ± SD DII	0.2 ± 1.8	−1.5 ± 1.0	1.6 ± 1.1	<0.0001
Mean ± SD Age (years)	47 ± 18	47 ± 17	46 ± 18	<0.0001
Mean ± SD Energy intake (kcal)	2130 ± 1004	2610 ± 1044	1719 ± 674	<0.0001
Mean ± SD Protein intake (g)	82 ± 43	102 ± 45	64 ± 29	<0.0001
Mean ± SD Systolic pressure (mmHg)	122 ± 18	121 ± 16	121 ± 17	0.6522
Mean ± SD Diastolic pressure (mmHg)	70 ± 13	71 ± 12	70 ± 12	0.0003
Mean ± SD Fasting glucose (mg/dL)	107 ± 32	104 ± 27	105 ± 29	0.0879
Mean ± SD Glycohemoglobin (%)	5.7 ± 1.0	5.6 ± 0.8	5.6 ± 1.0	<0.0001
Mean ± SD eGFR (ml/min/1.73 m^2^)	100 ± 17	99 ± 14	101 ± 16	<0.0001
Mean ± SD UACR (mg/g)	39 ± 301	22 ± 156	32 ± 254	0.0033
Gender (%)				<0.0001
Male	49	58	39	
Female	51	42	61	
Race (%)				<0.0001
Mexican American	15	9	9	
Other Hispanic	10	5	6	
Non-Hispanic White	44	72	64	
Non-Hispanic Black	21	8	14	
Other Race	11	7	7	
Hypertension (%)				0.1191
Yes	17.2	15.1	16.1	
No	76.2	84.9	83.9	
Diabetes (%)				<0.0001
Yes	13.3	8.9	11.4	
No	86.7	91.1	88.6	
CKD (%)				<0.0001
Yes	11.5	7.7	10.5	
CKD stage 1	5.6	3.8	5.4	
CKD stage 2	5.6	3.6	4.9	
CKD stage 3	0.3	0.2	0.3	
CKD stage 4/5	0	0	0	
No	88.5	92.3	89.5	
Physical activity (%)				<0.0001
Vigorous physical activity	35	45	33	
Moderate physical activity	32	33	33	
Less than moderate	33	22	34	
Smoking exposure (%)				<0.0001
Non-smoker	27	34	27	
Second-hand smoker	49	45	45	
Current smoker	24	21	28	
Alcohol intake (%)				<0.0001
Non-drinker	21	8	15	
Former drinker	12	8	13	
Current drinker	67	84	72	
BMI (%)				<0.0001
Normal (<25 kg/m^2^)	31	32	29	
Overweight (25–30 kg/m^2^)	32	36	30	
Obese (>30 kg/m^2^)	37	32	41	
All-cause mortality (%)	4.02	2.50	3.61	<0.0001

For categorical variables, the *p*-value was calculated by the weighted chi-square test. For continuous variables, the *p*-value was calculated by *t*-test. eGFR, estimated glomerular filtration rate; UACR, urinary albumin-to-creatinine ratio; CKD, chronic kidney disease; BMI, body mass index.

**Table 2 nutrients-14-02011-t002:** The relationship between dietary inflammatory index and diabetes.

Dietary Inflammatory Index (DII)	Model 1 ^a^	Model 2 ^b^	Model 3 ^c^
Fasting glucose (mg/dL)-β ^d^ (95% CI ^e^) *p*-value			
Continuous	0.45 (0.06, 0.84) 0.0236	0.83 (0.45, 1.22) < 0.0001	0.83 (0.30, 1.36) 0.0022
DII < 0	Ref = 0	Ref = 0	Ref = 0
DII > 0	1.54 (0.09, 2.98) 0.0370	2.72 (1.31, 4.14) 0.0002	1.91 (0.16, 3.66) 0.0323
Glycohemoglobin (%)-β ^d^ (95% CI ^e^) *p*-value			
Continuous	0.02 (0.02, 0.03) < 0.0001	0.03 (0.02, 0.04) < 0.0001	0.02 (0.01, 0.03) 0.0009
DII < 0	Ref = 0	Ref = 0	Ref = 0
DII > 0	0.07 (0.04, 0.10) < 0.0001	0.07 (0.04, 0.10) < 0.0001	0.03 (−0.01, 0.07) 0.1267
Diabetes-OR ^f^ (95% CI ^e^) *p*-value			
Continuous	1.07 (1.05, 1.10) < 0.0001	1.09 (1.06, 1.12) < 0.0001	1.05 (1.01, 1.09) 0.0139
DII < 0	Ref = 1	Ref = 1	Ref = 1
DII > 0	1.31 (1.19, 1.44) < 0.0001	1.34 (1.21, 1.49) < 0.0001	1.18 (1.03, 1.34) 0.0141
CKD-OR ^f^ (95% CI ^e^) *p*-value			
Continuous	1.10 (1.07, 1.13) < 0.0001	1.10 (1.06, 1.13) < 0.0001	1.14 (0.99, 1.31) 0.0660
DII < 0	Ref = 1	Ref = 1	Ref = 1
DII > 0	1.35 (1.22, 1.50) < 0.0001	1.31 (1.18, 1.46) < 0.0001	1.44 (0.93, 2.25) 0.1043
All-cause mortality-OR ^f^ (95% CI ^e^) *p*-value			
Continuous	1.08 (1.04, 1.13) 0.0003	1.12 (1.07, 1.17) < 0.0001	1.11 (1.03, 1.18) 0.0031
DII < 0	Ref = 1	Ref = 1	Ref = 1
DII > 0	1.33 (1.13, 1.56) 0.0007	1.44 (1.21, 1.72) < 0.0001	1.26 (1.02, 1.57) 0.0357

^a^ Model 1, unadjusted; ^b^ Model 2, adjusted for age, gender, and race; ^c^ Model 3, adjusted for age, gender, race, energy intake, protein intake, systolic pressure, diastolic pressure, smoking exposure, alcohol intake, and BMI; ^d^ β, regression coefficient; ^e^ CI, confidence interval; ^f^ OR, odds ratio. The results with statistical significance are shown in bold.

**Table 3 nutrients-14-02011-t003:** Baseline characteristics of participants with diabetes from NHANES 2009–2010, 2011–2012, and 2013–2014.

Dietary Inflammatory Index	Overall	Anti-Inflammatory Diet (DII < 0)	Proinflammatory Diet (DII > 0)	*p*-Value
Participant number	1904	811	1093	-
Mean ± SD DII	0.4 ± 1.8	−1.3 ± 0.9	1.6 ± 1.1	<0.0001
Mean ± SD Age (yrs)	60 ± 14	59 ± 14	60 ± 14	0.1376
Mean ± SD Energy intake (kcal)	1925 ± 907	2545 ± 951	1583 ± 652	<0.0001
Mean ± SD Protein intake (g)	79 ± 41	103 ± 42	63 ± 29	<0.0001
Mean ± SD Systolic pressure (mmHg)	132 ± 19	130 ± 18	131 ± 19	0.6375
Mean ± SD Diastolic pressure (mmHg)	69 ± 14	71 ± 14	69 ± 15	0.0020
Mean ± SD eGFR (ml/min/1.73 m^2^)	90 ± 14	89 ± 13	89 ± 13	0.5731
Mean ± SD UACR (mg/g)	148 ± 700	96 ± 452	138 ± 673	0.1179
Mean ± SD Fasting glucose (mg/dL)	155 ± 55	157 ± 55	153 ± 51	0.1978
Mean ± SD Glycohemoglobin (%)	7.6 ± 1.8	7.5 ± 1.7	7.5 ± 1.7	0.9987
Gender (%)				<0.0001
Male	54	67	42	
Female	46	33	58	
Race (%)				0.0012
Mexican American	17	10	10	
Other Hispanic	11	5	8	
Non-Hispanic White	37	66	58	
Non-Hispanic Black	24	11	16	
Other Race	11	8	9	
CKD (%)				<0.0001
Yes	29.5	23.2	28.9	
CKD stage 1	10.6	7.5	9.6	
CKD stage 2	17.8	14.6	18.4	
CKD stage 3	1.1	1.1	0.9	
CKD stage 4/5	0	0	0	
No	70.5	76.8	71.1	
Hypertension (%)				0.8896
Yes	30.5	29.6	29.9	
No	61.8	70.4	70.1	
Physical activity (%)				<0.0001
Vigorous physical activity	20	25	18	
Moderate physical activity	34	38	31	
Less than moderate	46	37	51	
Smoking exposure (%)				0.0002
Non-smoker	28	36	28	
Second-hand smoker	52	47	49	
Current smoker	20	17	23	
Alcohol intake (%)				<0.0001
Non-drinker	22	12	19	
Former drinker	15	10	17	
Current drinker	63	78	64	
BMI (%)				<0.0001
Normal (<25 kg/m^2^)	13	11	11	
Overweight (25–30 kg/m^2^)	28	26	25	
Obese (>30 kg/m^2^)	58	62	64	
All-cause mortality (%)	9.35	5.84	9.47	<0.0001

For categorical variables, the *p*-value was calculated by the weighted chi-square test. For continuous variables, the *p*-value was calculated by t-test. eGFR, estimated glomerular filtration rate; UACR, urinary albumin-to-creatinine ratio; CKD, chronic kidney disease; BMI, body mass index.

**Table 4 nutrients-14-02011-t004:** Relationship of dietary inflammatory index to all-cause mortality among diabetes patients (*n* = 178) carried out by Cox-regression analysis.

	Univariate Analysis		Multivariate Analysis	
	HR ^1^ (95% CI ^2^)	*p*-Value	HR^1^ (95% CI ^2^)	*p*-Value
DII > 0 (vs. DII < 0)	1.68 (1.23, 2.30)	0.0011	1.71 (1.13, 2.58)	0.0108
Female (vs. Male)	0.59 (0.44, 0.81)	0.0008	0.71 (0.49, 1.02)	0.0638
Non-Hispanic White (vs. Other races ^3^)	2.02 (1.27, 3.21)	0.0031	0.96 (0.58, 1.60)	0.8739
Age	1.07 (1.05, 1.08)	<0.0001	1.07 (1.04, 1.09)	<0.0001
Energy intake	0.99 (0.99,1.01)	0.0802	0.99 (0.99, 1.00)	0.0056
Protein intake	0.99 (0.98,1.00)	0.0128	0.99 (0.98, 1.02)	0.2423
eGFR ^4^	0.96 (0.95,0.97)	<0.0001	0.98 (0.95, 1.00)	0.0601
UACR ^5^	1.00 (1.00, 100)	<0.0001	1.00 (0.99, 1.00)	0.0023
Systolic pressure	1.01 (1.00,1.02)	0.0096	0.99 (0.98, 1.00)	0.1786
Diastolic pressure	0.98 (0.97,0.99)	0.0124	1.00 (0.99, 1.01)	0.4671
Non-drinker (vs. Current drinker)	0.86 (0.60, 1.26)	0.4510	1.27 (0.83, 1.94)	0.7722
Former drinker (vs. Current drinker)	1.12 (0.75, 1.66)	0.5760	0.94 (0.60, 1.45)	0.2740
Moderate physical activity (vs. Less than moderate)	0.57 (0.41, 0.80)	0.0011	0.73 (0.51, 1.03)	0.0778
Vigorous physical activity (vs. Less than moderate)	0.40 (0.24, 0.66)	0.0003	0.60 (0.34, 1.02)	0.0576
Obese (vs. Normal)	0.56 (0.37, 0.85)	0.0059	0.85 (0.54, 1.33)	0.4673
Non-smoker (vs. Current smoker)	0.11 (0.07, 0.18)	<0.0001	0.06 (0.04, 0.11)	<0.0001
Second-hand smoker (vs. Current smoker)	0.63 (0.40, 0.99)	0.0442	0.39 (0.24, 0.66)	0.0003

^1^ HR, hazard ratio; ^2^ 95% CI, 95% confidence interval; ^3^ Other races, including Mexican American, Other Hispanic, Non-Hispanic Black, and others; ^4^ eGFR, estimated glomerular filtration rate; ^5^ UACR, urinary albumin-to-creatinine ratio. Results with statistical significance are emboldened.

## Data Availability

All data in our study are publicly available at https://www.cdc.gov/nchs/nhanes/ (accessed on 1 January 2022).

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
