# Peer review of "A Proinflammatory Diet May Increase Mortality Risk in Patients with Diabetes Mellitus"

_nutrients, 2022, doi:10.3390/nu14102011_

Round 1

Reviewer 1 Report

I enjoyed reviewing this interesting manuscript. The article is methodologically correct. The conclusions are supported by the results. This reviewer only gives a few suggestions

1- In Conclusion the authors state “…more clinical trials are needed to help clarify the exact effect of the proinflammatory diet.” Intriguingly, a study showed that high-fat meal produces further increase in TNF-alpha levels associated with endothelial dysfunction (Nutr Metab Cardiovasc Dis. 2007 May;17(4):274-9. doi: 10.1016/j.numecd.2005.11.014.). This point should be focused in discussion.

2- A linguistic revision by a native English speaker is required.

Author Response

Please find the response in attachment

Reviewer 2 Report

This retrospective analysis, based on merging two large US-based databases (NHANES and NCI), analyzed the association between self-reported pro-inflammatory diet and all-cause mortality in 15,291 patients and a subset of 1904 patients with a convincing history of diabetes mellitus (DM). The study timeline was five years 2009-2014. The number of patients who consumed an anti-inflammatory diet (DII<0) was fairly well matched to the number who had consumed a pro-inflammatory diet (DII>0).     

     Primary endpoint was risk (hazards ratio, HR) of mortality by all causes, with secondary endpoints the HR for fasting hyperglycemia, elevated glycosylated Hb (HbA1c), incidence of DM  and renal disease. Statistical analysis consisted of Chi-Square or t-test, and univariate and multivariate  Cox linear regression to define HR. In the linear regression analysis, three models were utilized to adjust for co-variates.

Authors found, in the larger inclusive sample, that pro-inflammatory diet was associated with increased risk of DM.  The statistically significant positive association with presence of DM (11.4 vs 8.9%) and CKD (10.5 vs 7.7%) and all-cause mortality (3.6 vs 2.5%) are noteworthy,  while those with fasting hyperglycemia (105 vs 103 mg/dl) and increased HbA1c (5.64 vs 5.57%), while statistically significant, are not clinically important. Risk analysis using their most useful Model 3, indicated there was a 18% higher risk of DM in the pro-inflammatory diet group. Risk of the CKD and all-cause mortality in all comers, unfortunately, was not reported. When restricting scope to patients DM, pro-inflammatory diet was associated with decreased caloric intake (1583 vs 2545 kcal), lower protein intake (63 vs 103 g), female sex (58% vs 33%), low physical activity (51 vs 37%),  increased incidence CKD (29 vs 23%) and all-cause mortality (9.5 vs 5.8%). There was no connection between diet and incidence of obesity in the diabetic subset. Association of diet with hypertension diagnosis, and markers of glycemic control (fasting glucose, HbA1c), unfortunately, was not reported. In Table 4, the pro-inflammatory diet presented a 1.71 risk of  all-cause mortality in the multivariate linear regression analysis. Study did not report risk factors for glycemic control (either fasting glucose or HbA1c), CKD or hypertension) which should have been possible given the datasets available for investigation.

Critique: There are a number of gaps in the reporting, as outlined in the previous paragraph. These would have been helpful to sort out the relationships between diet and variables glucose control and renal injury (which is known to be complication of DM). With respect to Tables, reporting values should be restricted to whole numbers with exceptions for DII, HbA1c, % disease incidence (which all should be 0.1 value) and all-cause mortality (which can remain as 0.01 value).  It is not clear from the methods the duration of the diet as self-reported by the subject, and this should either be reported or discussed as weakness of the study.

 In Table 1 and lines 163-166, the diet association with disease is clinically important for CKD and DM, but small magnitude makes it clinically unimportant for HbA1c, GFR, UACR. In fact, there was no difference in incidence of hypertension (16 vs 15%), which should be reported as a line item in Table 1.  Similarly in lines 182-184, these statistical differences on glycemia and HbA1c are far too small to have clinical implication and should be stated as such. Table 3 is missing line items for incidence of hypertension, glycemia and HbA1c and  as well.  

Other weaknesses that authors must emphasize: 1) association does not imply cause-and- effect relationship , 2) unknown duration and true accuracy of dietary intake, and 3) inter-relationship between comorbid factors on mortality is not possible to evaluate in study design.

Linguistic corrections:

Line 39: Replace “are suffering” with “suffer”

Line 40: Omit “Apparently”

Line 48: Omit “cause and”

Line 52: Omit “Subsequently”

Line 58: Replace “preliminary” with “primary”

Line 72: Omit “primarily”

Lines 91-98: Omit all that refers to the technique for measuring glucose, as this was NOT performed by the present investigators who only had access to published data.

Lines182-183: Must state that the numerical difference for glucose or HbA1c, while statistically capable of achieving definition of “different” are far too small to matter to a clinician or patient.

Line 197: Replace “recruited” with “evaluated”. Authors did not recruit any subjects for study.

Line 199: Omit “significant”

Round 2

Reviewer 2 Report

This revised manuscript, based on merging two large US-based databases (NHANES and NCI), analyzed the association between self-reported pro-inflammatory diet and all-cause mortality in 15,291 patients and a subset of 1904 patients with a convincing history of diabetes mellitus (DM). The study timeline was five years 2009-2014. The number of patients who consumed an anti-inflammatory diet (DII<0) was fairly well matched to the number who had consumed a pro-inflammatory diet (DII>0).     

     Primary endpoint was risk (hazards ratio, HR) of mortality by all causes, with secondary endpoints the HR for fasting hyperglycemia, elevated glycosylated Hb (HbA1c), incidence of DM  and renal disease. Statistical analysis consisted of Chi-Square or t-test, and univariate and multivariate  Cox linear regression to define HR. In the linear regression analysis, three models were utilized to adjust for co-variates.

Authors responded well to previous criticism, including further analysis of the diabetic subset, which showed that the pro-inflammatory diet was neither associated with worse control of diabetes mellitus (same Hb1Ac 7.5% concentration as low-inflammatory diet group) nor higher incidence of hypertension (30% in both groups). Risk of CKD and all-cause mortality in all comers, now added, show pro-inflammatory diet associated with 26-44% increased risk of all-cause mortality, depending on model. Similar numbers for CKD were only statistically different in two of the three models employed. Authors have filled the previous reporting gaps and clarified presentation in the tables. Lastly, authors have now more completely acknowledged weaknesses of the design study, from which they have now extracted the maximum potential.

Authors adopted this reviewer’s previous linguistic corrections. At present I have only a few remaining editorial housekeeping comments:

Line 333: Replace “significant” with “material”

Line 334: Sentence should read:  “The observed differences in plasma concentrations of fasting glucose and glycohemoglobin are too small to be considered clinically relevant”.

Line 350: Sentence should read: “There was no difference in prevalence of hypertension….”

Line 551: Omit “significant”

Line 617: Replace “however” with “although”

Line 1003: Sentence should read: “Esposito et al., in 25 patients with metabolic syndrome, found that high-fat meal…”

Lines 1022: Omit “thus”

Line 1039-41: Sentences should read “… no causal relationship could be established between DII and prevalence of diabetes mellitus. Association does not imply a cause-and-effect relationship. Prospective longitudinal investigation will be required to explore that matter.”

Lines 1095: Replace “debatable” with “uncertain”
